# A Non-Gradual Development Process of Cicada Eyes at the End of the Fifth-Instar Nymphal Stage to Obtain Visual Ability

**DOI:** 10.3390/insects13121170

**Published:** 2022-12-16

**Authors:** Minjing Su, Feimin Yuan, Tiantian Li, Cong Wei

**Affiliations:** Key Laboratory of Plant Protection Resources and Pest Management, The Ministry of Education, College of Plant Protection, Northwest A&F University, Xianyang 712100, China

**Keywords:** visual organ, development, pigment, phototransduction, adaptation, transcriptome

## Abstract

**Simple Summary:**

Visual systems of insects reflect high adaptation to their habitats. Cicadas undergo a significant niche change during ontogenetic development, which results in an obvious difference in color and ultrastructure of the compound eyes between nymphal and adult stages. However, little is known about when and how cicadas obtain their visual ability to deal with the new habitat. Here we use transcriptome analyses to clarify the timing and potential molecular mechanisms underlying the structural and functional transformation of the eyes of the cicada *Meimuna mongolica*. These results improve understanding of developmental and adaptive changes in compound eyes in insects which undergo a drastic ontogenic shift in habitat.

**Abstract:**

Insects’ visual system is directly related to ecology and critical for their survival. Some cicadas present obvious differences in color and ultrastructure of compound eyes between nymphal and adult stages, but little is known about when cicadas obtain their visual ability to deal with the novel above-ground habitat. We use transcriptome analyses and reveal that cicada *Meimuna mongolica* has a trichromatic color vision system and that the eyes undergo a non-gradual development process at the end of the 5th-instar nymphal stage. The white-eye 5th-instar nymphs (i.e., younger 5th-instar nymphs) have no visual ability because critical components of the visual system are deficient. The transformation of eyes toward possessing visual function takes place after a tipping point in the transition phase from the white-eye period to the subsequent red-eye period, which is related to a decrease of Juvenile Hormone. The period shortly after adult emergence is also critical for eye development. Key differentially-expressed genes related to phototransduction and chromophore synthesis play positive roles for cicadas to adapt to above-ground habitat. The accumulation of ommochromes corresponds to the color change of eyes from white to red and dark brown during the end of the 5th-instar nymphal period. Cuticle tanning leads to eye color changing from dark-brown to light-brown during the early adult stage. We hypothesize that the accumulation of ommochromes occurring at the end of 5th-instar nymphal stage and the early adult stage is not only for cicadas to obtain visual ability, but also is a secure strategy to cope with potential photodamage after emergence.

## 1. Introduction

Insecta is the most diverse class of animals, having evolved to occupy various ecological niches from the brightest to the darkest [1]. Most insects have some sight, except for a few subterranean and endoparasitic species [2]. Effective vision is vital to many insects, with highly-developed visual systems critical for survival [3]. The compound eye is the main visual organs of insects, enabling them to recognize their environment efficiently [4].

Compound eyes consist of a number of ommatidia consisting of several parts, i.e., corneal lens, crystalline cone, rhabdom, and a group of photoreceptors and pigment cells [5]. Molecules involved in the phototransduction cascade of ommatidia convert light into electrical signals [6]. Pigments existing in pigment cells are closely related to the color of compound eyes, and critical for the optical function [7,8] and retinal protection [9,10,11]. Any slight change of any of the structures, molecules and pigments of compound eyes could generate significant variations in vision, which in turn is directly related to transformations in behavior, ecology and adaptation. For example, the halictid bee *Megalopta genalis* with apposition eyes, which are suited for vision in bright light in terms of the structure [1], is a nocturnal active flyer with eyes that are structurally modified (i.e., the diameter of lens and rhabdom being increased) to facilitate its operation under low light conditions [12].The American cockroach *Periplaneta americana* with apposition eyes also adapts well to dark environments, due to that the expression of TRPL (transient receptor potential-like channel) is more abundant than TRP (transient receptor potential channel) [13]. The type and abundance of opsins are closely related to the adaptation of visual environment in moths [14,15], dragonflies [16] etc. The compound eyes with different color in triatomine bugs presents different degrees of damage when they are exposed to ultraviolet [10].

Cicadas (Hemiptera: Cicadoidea) undergo significant niche changes during ontogenetic development (Figure 1). They are exclusively subterranean during their long nymphal stage and then emerge to only have a short adult lifespan above ground [17]. Their light environment changes dramatically from dark to bright during their development from the nymphal stage to adult. A previous study revealed that the compound eyes of 5th-instar nymphs and adults of the cicada *Meimuna mongolica* (Distant) exhibit significant differences [18] (Figure 1). For example, the color of the eyes of the 5th-instar nymphs gradually changes from white to red and dark brown before the adult emergence. The eyes of the adults are light brown. Different types of sensilla exist on the surface of eyes in the 5th-instar nymphal phase, which disappear in the adult phase. The white eyes of nymphs have a complete cornea and a large amount of clustered cells but lack visible ommatidia. The red and dark brown eyes of nymphs are composed of many irregular, hexagonal or pentagonal ommatidia whereas the eyes of adults are composed of equilateral hexagonal ommatidia [18]. These morphological differences indicate that the major developmental period of compound eyes of cicadas may occur at the end of the final instar nymphal stage [18].

Here the compound eyes of *M. mongolica* are selected as a model to further investigate the eye development of cicadas. Transcriptome analyses were used to clarify the potential molecular mechanisms underlying the structural and functional changes of cicada eyes during the period before and after adult emergence. This study is based on RNA-Sequencing data of *M. mongolica* eyes from different developmental stages/samples. This includes the 5th-instar nymphs with white eyes (N5W), with red eyes (N5R) and with dark brown eyes (N5B) and also the adults that emerged from the earth for 3 h (A3) and 48 h (A48), respectively. To define the timing of structural and functional changes of eyes, hierarchical cluster analysis and expression profiles were performed based on all identified differentially-expressed genes (DEGs). Subsequently, functional enrichment analyses were performed based on both stage-specific DEGs and profile-specific DEGs. Then we focused on key DEGs to define the critical timing of eye changes and to clarify the potential molecular mechanism underlying the eye development. These results may improve our understanding of developmental and adaptive changes of compound eyes in insects which undergo a drastic ontogenic shift in habitat.

## 2. Materials and Methods

### 2.1. Sample Preparation

All individuals of *M. mongolica* were obtained from the campus of Northwest A & F University, Yangling, Shaanxi Province, China during the period from 20 July to 5 August 2020. The nymphs were captured by digging the soil beneath the host plants (*Pyrus xerophila* T. T. Yu). The head capsule width of nymphs was measured in the laboratory of Northwest A&F University immediately after their capture. The 5th-instar nymphs were selected according to their head capsule width [20] together with the color of their compound eyes [18]. The newly-emerged adults were captured on the trunk of the host plants at night, approximately at 22:00–24:00. The emerged cicadas were maintained at 25 ± 1 °C under relative humidity 50 ± 5% with a photoperiod of 14 h light/10 h dark. They were reared on branches of the host plants in the laboratory. Adults were collected after emerging from their exuviae at 3 and 48 h, respectively.

The study samples involve five development stages, i.e., the 5th-instar nymphs with white eyes (N5W), with red eyes (N5R), with dark brown eyes (N5B), and adults that had emerged from exuviae for 3 h (A3) and 48 h (A48), respectively. The compound eyes (including lens, retina and optic lobes) of all samples were removed and immediately flash-frozen in liquid nitrogen, and were stored at −80 °C for subsequent RNA extraction. The eyes of two individuals were randomly selected and pooled as one biological sample, and three biological replicates for each sampled stage (N5W, N5R, N5B, A3 and A48) were selected, resulting in 15 (3 × 5) samples.

### 2.2. RNA Extraction, Raw Data Quality Control and cDNA Library Construction

Trizol reagent (Invitrogen, Carlsbad, CA, USA) was used for the extraction of Total RNA following the manufacturer’s procedure. Analyses of the total RNA quantity and purity were based on Bioanalyzer 2100 and RNA 1000 Nano LabChip Kit (Agilent, Santa Clara, CA, USA) (with an RIN number > 7.0). The poly(A) RNA was purified from total RNA (5 µg) using poly-T oligo-attached magnetic beads, and using two rounds of purification. After the purification, the mRNA was fragmented into small pieces using divalent cations under elevated temperature. The cleaved RNA fragments were then reverse-transcribed to create the final cDNA library in accordance with the protocol for the mRNA Seq sample preparation kit (Illumina, San Diego, CA, USA). The average insert size for the paired-end libraries was 300 bp (±50 bp). Subsequently, we performed the paired-end sequencing on the IlluminaHiseq4000 (LC Sciences, Houston, TX, USA) following the vendor’s recommended protocol.

### 2.3. De Novo Assembly, Unigene Annotation and Functional Classification

Cutadapt [21] was used to remove the reads containing adaptor contamination, low quality bases and undetermined bases. The quality of sequence was verified using FastQC (http://www.bioinformatics.babraham.ac.uk/projects/fastqc/ (accessed on 24 September 2020)), including the Q20, Q30 and GC-content of the clean data. All downstream analyses were based on clean high-quality data. Trinity 2.4.0 [22] was used for the de novo assembly of the transcriptome. Trinity groups transcripts into clusters based on shared sequence content. Such a transcript cluster is very loosely referred to as a ‘gene’. The longest transcript in the cluster was selected as the gene sequence (aka Unigene).

All assembled unigenes were aligned against five databases using DIAMOND [23] with a threshold of E-value < 0.00001, including the non-redundant (Nr) protein database (http://www.ncbi.nlm.nih.gov/ (accessed on 25 September 2020)), Gene ontology (GO) (http://www.geneontology.org (accessed on 25 September 2020)), SwissProt (http://www.expasy.ch/sprot/ (accessed on 25 September 2020)), Kyoto Encyclopedia of Genes and Genomes (KEGG) (http://www.genome.jp/kegg/ (accessed on 25 September 2020)) and eggNOG (http://eggnogdb.embl.de/ (accessed on 25 September 2020)). In addition, CutProtFam-Pred [24] was used for the further classification of putative structural cuticular proteins from sequence alone.

### 2.4. Analysis of Differentially Expressed Genes (DEGs)

Salmon [25] was applied to determine expression levels for unigenes by calculating TPM (transcripts per million) [26]. The DEGs were selected with log2 (fold change) > 1 or log2 (fold change) < −1, statistical significance (*p*-value < 0.01) and false discovery rate-adjusted P (FDR/q-value) < 0.01 using edgeR package [27] in R (version 4.1.1) (R Foundation for Statistical Computing, Vienna, Austria). DEGs of *M. mongolica* compound eyes were identified in the comparable groups of N5W vs. N5R, N5W vs. N5B, N5W vs. A3, N5W vs. A48, N5R vs. N5B, N5R vs. A3, N5R vs. A48, N5B vs. A3, N5B vs. A48, and A3 vs. A48, respectively. 

### 2.5. Fuzzy C-Means Clustering

All DEGs were grouped into 10 different clusters (profiles) using Mfuzz package [28] in R (version 4.1.1) (R Foundation for Statistical Computing, Vienna, Austria) with fuzzy c-means algorithm.

### 2.6. Enrichment Analysis of GO Enrichment and KEGG Pathway

GO terms and KEGG pathways enrichment analysis were performed on the DEGs of the comparable groups of N5W vs. N5R, N5R vs. N5B, N5B vs. A3, A3 vs. A48, and the 10 profiles respectively. Functional enrichment analyses were carried out using the clusterProfiler package [29] in R (version 4.1.1) (R Foundation for Statistical Computing, Vienna, Austria), and the gene sets with *p*-value < 0.01 and Fold Enrichment > 5 were focused. 

### 2.7. Phylogenetic Analysis

The four opsin genes were aligned with corresponding genes of related species to ensure the reliability of the four genes as the homologs of opsin genes. All selected nucleotide sequences were aligned using the ClustalW tool in MEGA-7 software [30] before phylogenetic analysis. Phylogenetic analyses are based on 29 nucleotide sequences, including the four opsin genes from *M. mongolica* identified in this study, 24 opsin genes from seven other insect species and one opsin gene from the squid *Todarodes pacificus*. Subsequently, the phylogenetic analyses were conducted using MEGA-7 software [30] with the neighbor-joining (NJ) and maximum likelihood (ML) methods, and bootstrap values were obtained based on 500 bootstrap replications. The names and sequence numbers of four opsin sequences in *M. mongolica* are listed in Appendix A. In addition, protein sequences of the 29 opsins were aligned using the ClustalW tool, and transmembrane domains were identified using the TMHMM Server (Version 2.0, http://www.cbs.dtu.dk/services/TMHMM/, accessed on 11 December 2022).

### 2.8. Reverse-Transcription Quantitative PCR (RT-qPCR)

To verify the quantification of gene expression levels in the RNA-sequencing, qPCR was performed for eight genes in *M. mongolica*. Their primer sequences are given in Appendix A, and Primer-BLAST (NCBI) (https://www.ncbi.nlm.nih.gov/tools/primer-blast/index.cgi, accessed on 10 September 2021) was used for the confirmation of their specificity. For RT-qPCR experiments, independent total RNA samples were extracted using the same procedure as for the RNA-Seq experiments and reverse-transcribed to cDNA using the Hifair III 1st Strand cDNA Synthesis Kit (Yeasen, Shanghai, China). Relative expression levels were analyzed using the LightCycler 480 system (Roche, Basel, Switzerland). The Real-Time PCR Detection System (ABI 7500) (Applied Biosystems, Foster, CA, USA) was used for the quantitative reaction. The reaction mixture (20 μL) contained 2× T5 (syber green) 10 μL, 1 μL each of the forward and reverse primers, 1 μL of template cDNA, and 7 μL of ddH_2_O. The PCR temperature profile was set to 95 °C for 1 min, followed by 40 cycles of 95 °C for 15 s and 56 °C for 15 s, with melt curves stages at 95 °C for 5 s, 61 °C for 1 min, and 95 °C for 15 s. After amplification, an analysis of the melting curve was used for the detection of a single gene-specific peak and checking of the nonspecific amplification. Negative controls without the template were contained in each experiment. All qRT-PCR experiments were repeated in three biological and three technical replications. The 2^−ΔΔCt^ method [31] was used for the determination of fold changes. The reference gene was the 50S ribosomal protein gene, which was used as a normalizer gene for *M. mongolica*. The SPSS Statistics 21.0 software (IBM, Chicago, IL, USA) was used for data analyses. Results are reported as mean ± SE.

### 2.9. Accession Number

The RNA-Seq has been submitted to the Sequence Read Archive (SRA) of the National Center for Biotechnology Information under BioProject number PRJNA888061. The release date is 14 October 2022.

## 3. Results

### 3.1. Stage-Specific RNA-Seq of Compound Eyes of M. mongolica

The depth of sequencing we carried out was >6 GB per mixed sample (including two individuals for each repetition of the five sampled stages, i.e., N5W, N5R, N5B, A3, A5) to ensure coverage of the *M. mongolica* transcriptomes. The number of obtained raw reads of transcriptome sequencing of 15 samples (5 sampled stages × 3 biological replicates) was 721,835,216. In total, 712,483,158 valid reads were acquired after filtering, and the valid reads per sample ranged from 38 to 57 million. The valid reads were de novo assembled into 46,399 unigenes, with their length ranging from 201 to 10,362 bases (median length = 331 bases, and N50 = 1034 bases). All assembled unigenes were aligned to five authoritative databases: (the non-redundant (Nr) protein database, Gene ontology (GO), SwissProt, Kyoto Encyclopedia of Genes and Genomes (KEGG), and eggnog databases) using DIAMOND with a threshold of Evalue < 0.00001.

### 3.2. Cluster Analyses Based on All Identified Differentially-Expressed Genes

In total, 7431 differentially-expressed genes (DEGs) of *M. mongolica* were identified. Results of unsupervised hierarchical clustering based on all the identified DEGs for clarifying the overall difference of gene expression in different samples showed that all the samples of 5th-instar nymphs (i.e., N5W, N5R and N5B) were clustered into one group, and the remaining two samples of adults (i.e., A3 and A48) were clustered into the other group (Appendix A). This suggests that drastic changes in eyes occurred during the period between N5B and A3, corresponding to the nymph-adult transition and the niche change. In addition, all samples of 5th-instar nymphs were clustered into two independent subgroups, with one subgroup containing the samples of N5W and the other group containing the samples of N5R and N5B (Appendix A). This implies that the period between N5W and N5R is a critical stage in eye development and corresponds to the eye color change from white to red during the nymphal stage.

To further understand the dynamic changes of the expression levels of DEGs during development, cluster analysis of expression profiles was performed on all DEGs. In total, 10 distinct profiles of temporal dynamics were observed. Among these profiles, Profile 4 represented DEGs being down-regulated, and Profiles 2, 5 and 9 represented DEGs being up-regulated, for which significant changes in expression level appeared after a point in time but not gradually. Profiles 1, 3, 6–8 and 10 represented DEGs displaying multimodal expression patterns, which also indicate this development of eyes is not gradual but underwent drastic changes at certain time points. In detail, significant changes of the expression of DEGs were revealed during the period between N5B and A3 (Profiles 1–3 and 8–10) and the period between N5W and N5R (Profiles 3, 4 and 6–8) which suggest that the tipping point or critical stage for eye development was during these two periods.

### 3.3. Functional Enrichment Analysis Based on Stage-Specific and Profile-Specific DEGs 

Different expression analyses detected 2309 DEGs between N5W and N5R, 890 DEGs between N5R and N5B, 3260 DEGs between N5B and A3, and 1594 DEGs between A3 and A48. GO function and KEGG pathway enrichment analyses were performed on the stage-specific DEGs, which resulted in 72 gene sets (50 GO terms and 22 KEGG pathways) between N5W and N5R, 27 gene sets (18 GO terms and nine KEGG pathways) between N5R and N5B, 68 gene sets (49 GO terms and 19 KEGG pathways) between N5B and A3, and 38 gene sets (28 GO terms and ten KEGG pathways) between A3 and A48 (all with *p*-value < 0.01). We focused on GO terms and KEGG pathways with Enrichment Fold > 5 to reveal the primary biological functions of related DEGs. Seventeen gene sets (13 GO terms and four KEGG pathways) between N5W and N5R, 19 gene sets (14 GO terms and five KEGG pathways) between N5R and N5B, four gene sets (four GO terms) between N5B and A3, and 14 gene sets (11 GO terms and three KEGG pathways) between A3 and A48 met the screening conditions. Functions of these gene sets were mainly involved in five aspects, including coloration, phototransduction, hormonal regulation, innate immune response, and cuticle development (Appendix A), among which the gene sets involved in coloration and phototransduction have a direct influence on light sensitivity and vision. The gene sets related to coloration were enriched in three comparable groups (N5W vs. N5R, N5R vs. N5B, and A3 vs. A48). The gene sets related to phototransduction were abundant in two comparable groups (N5W vs. N5R and N5R vs. N5B). The gene sets related to hormonal regulation or innate immune response were abundant in three comparable groups (N5W vs. N5R, N5R vs. N5B, and A3 vs. A48). Moreover, the gene sets related to cuticle development were abundant in all comparable groups.

Subsequently, GO function and KEGG pathway enrichment analyses were performed on DEGs in the 10 profiles, respectively. The significantly-enriched gene sets (*p*-value < 0.01, Enrichment Fold > 5) were targeted and analyzed. These gene sets are related to specific biological functions including coloration, phototransduction, hormonal regulation, innate immune response and cuticle development, respectively (Figure 2).

### 3.4. DEGs Related to Pigmentation

Two gene sets related to “tryptophan metabolism (map00380)” and “tyrosine metabolism (map00350)” were enriched in Profile 4 (Figure 2), which are involved in pigment biosynthesis in pigment cells of compound eyes and melanization of the cuticle, respectively. Changes in the expression levels of the key DEGs could be related to the color change of compound eyes; thus, these DEGs were targeted in the following analyses. In contrast, no genes related to pteridine biosynthesis pathway were found, indicating that pteridines do not exist in the cicada eyes. 

In regard to the ommochrome biosynthesis pathway (Figure 3), the expression level of TDO (TRINITY_DN34737_c0_g1), which encodes enzyme tryptophan 2,3-dioxygenase, was extremely low in N5W. This indicates that there was no accumulation of ommochrome in N5W, which could lead to the compound eyes being white in N5W as well as in the younger nymphal stages. The expression level of three genes, i.e., TDO and KMO (TRINITY_DN35565_c3_g1) encoding enzyme kynurenine 3-monooxygenase, and scarlet (TRINITY_DN26544_c0_g4) encoding an ABC transporter, presented a significant uptrend from N5W to N5R (Figure 3 and Figure 4). The expression level of four genes, i.e., TDO, KMO, scarlet and white (TRINITY_DN31752_c0_g2) encoding another ABC transporter presented a significant downtrend from N5B to A3 (Figure 3 and Figure 4). This suggests that continuous accumulation of ommochromes was started at the end of N5W, which could contribute to the gradual color change of compound eyes from white to red and dark brown. In addition, the expression levels of KMO and scarlet were significantly up-regulated in A48 when compared with A3 (Figure 3 and Figure 4), indicating that these two genes could be involved in oxidation resistance of ommochrome at the adult stage. 

In regard to the melanin biosynthesis pathway (Figure 3), cuticular-secreted protein genes were mainly expressed in the 5th-instar nymphal stage. Specifically, the expression level of the gene yellow (TRINITY_DN24657_c0_g2) encoding the protein yellow in N5W was the highest; the expression level of the gene yellow (TRINITY_DN29177_c0_g1) in N5R was the highest; and the expression level of the gene yellow (TRINITY_DN31457_c0_g1) in N5R and N5B was the highest. The gene (TRINITY_DN28647_c0_g1) encoding the enzyme laccase (Lac2) was highly-expressed in N5B, and the other gene (TRINITY_DN38674_c0_g1) encoding Lac2 was highly-expressed in N5R. In addition, the epidermal enzyme gene DDC (TRINITY_DN32493_c1_g1) encoding DOPA decarboxylase was highly-expressed in N5W and N5B (Figure 3). The other three epidermal enzyme genes, i.e., tan (TRINITY_DN24719_c0_g10) encoding NBAD hydrolase, ebony (TRINITY_DN32496_c0_g5) encoding NBAD synthase, and TH (TRINITY_DN31278_c0_g2) encoding tyrosine hydroxylase, were mainly expressed in the adult samples (Figure 3). Specifically, TH, tan and ebony all exhibited a dramatic up-regulated trend from N5B to A3; TH reached a peak in A3 and exhibited a dramatic downtrend from A3 to A48; and tan and ebony had no significant difference from A3 to A48 (Figure 3 and Figure 4). The expression patterns of the above-mentioned genes, in particular TH, indicate that the eyes were tanned mainly after adult emergence and that the melanization of eyes did not occur at the 5th-instar nymphal stage.

### 3.5. DEGs Related to Structure of Visual System

Four gene sets related to visual bases were revealed in Profiles 2, 5 and 7. The expression level of these DEGs in N5W was the lowest and in A48 the highest (Figure 2). Two gene sets related to light sensitivity of compound eyes, i.e., “photoreceptor cell maintenance (GO:0045494)” and “phototransduction (GO:0007602)”, were enriched in Profile 2, which represent DEGs displaying a significant uptrend from N5B to A3. This indicates that the adult emergence period, in addition to the period between N5W and N5R, is a critical point in time for terminal differentiation of retinal development. Two gene sets related to rhodopsin, “G-protein coupled receptor (GO:0001664)” and “retinol metabolism (map00830)”, were enriched in Profile 5 and Profile 7, respectively, which represent DEGs both displaying a dramatic uptrend from A3 to A48. In addition, the gene set “rhabdomere (GO:0016028)” related to construction of compound eyes was enriched between N5W and N5R, indicating that the eye development and functional transformation could be started during this period.

Key DEGs involved in phototransduction cascade, visual cycle and ocular retinoid metabolism were subsequently selected to analyze the process of development and functional transformation of compound eyes in detail (Figure 5). As a result, nine key DEGs involved in phototransduction cascade were revealed: TRP (TRINITY_DN29660_c0_g3) encoding Ca^2+^-permeable light-sensitive transient receptor potential channel, TRPL (TRINITY_DN30385_c2_g5) encoding transient receptor potential-like channel, NinaC (TRINITY_DN28560_c2_g3) encoding myosin III, two genes (i.e., TRINITY_DN28826_c1_g1, TRINITY_DN29311_c0_g2) encoding protein Arrestin homologs, and four genes (i.e., TRINITY_DN29661_c2_g1, TRINITY_DN29661_c2_g2, TRINITY_DN25451_c0_g4, and TRINITY_DN25583_c0_g4) encoding respectively opsins MmLop1, MmLop2, MmBLop1 and MmUVop1 (Figure 5). The expression levels of these DEGs in N5W were the lowest and in A3 or A48 the highest (Figure 5), indicating the nymphs with white eyes have no visual ability and radical changes related to eye development and functional transformation occur in the late 5th-instar nymphal stage. The first intense uptrend of these DEGs appeared in the period from N5W to N5R (Figure 4 and Figure 5), which indicates that the tipping point of eye development exists in this period. A subsequent drastic rise of expression of eight of these DEGs (except for NinaC) occurred from N5B to A3 (Figure 4 and Figure 5). In addition, three of these DEGs were significantly up-regulated from A3 to A48, i.e., MmUVop1, TRP and Arr (TRINITY_DN28826_c1_g1) (Figure 4 and Figure 5). The expression patterns of these DEGs confirm that the period of adult emergence is a critical point in time for functional development of the phototransduction cascade of eyes.

Eight DEGs involved in chromophore de novo synthesis and regeneration were revealed (Figure 5). They are as follows: two genes (TRINITY_DN31704_c0_g4, TRINITY_DN28149_c3_g3) encoding retinoid-binding protein (Pinta), two genes (TRINITY_DN27205_c0_g1, TRINITY_DN27205_c0_g4) encoding enzyme β, β-carotene-15,15′monooxygenase (NinaB), and four genes (TRINITY_DN34602_c0_g4, TRINITY_DN30505_c1_g2, TRINITY_DN26316_c0_g6, and TRINITY_DN34823_c0_g1) encoding photoreceptor dehydrogenase (PDH). In general, the expressions of these genes were low in N5W, which indicates that chromophore is not synthesized in the 5th-instar nymphs with white compound eyes (i.e., the nymphs with white eyes had no visual ability). The expression levels of four genes presented a dramatic uptrend from N5W to N5R (Figure 4 and Figure 5), including the two NinaB genes, and the two genes (TRINITY_DN34602_c0_g4, TRINITY_DN26316_c0_g6) encoding PDH, which confirms that the tipping point of eye development appears in the period from N5W to N5R. The expression levels of two genes displayed a significant uptrend from N5R to N5B (Figure 4 and Figure 5), including one gene (TRINITY_DN27205_c0_g4) encoding NinaB and one gene (TRINITY_DN34823_c0_g1) encoding PDH. The expression levels of four genes displayed a significant uptrend from N5B to A3 (Figure 4), including the two NinaB genes, the one gene (TRINITY_DN28149_c3_g3) encoding Pinta and one gene (TRINITY_DN34823_c0_g1) encoding PDH. The expression levels of three genes presented a significant uptrend from A3 to A48 (Figure 4 and Figure 5), including the three genes (TRINITY_DN30505_c1_g2, TRINITY_DN26316_c0_g6, and TRINITY_DN34823_c0_g1) encoding PDH.

To confirm the types of the four opsins in *M. mongolica*, the four opsin genes were aligned with corresponding genes of closely related species, and a maximum likelihood (ML) tree and a neighbor-joining (NJ) tree were constructed. These two trees showed the same topology (Figure 6 and Appendix A). Results of phylogenetic analyses of these opsin genes showed that all the genes from insects formed three well-supported clades, and their encoding opsins were separately sensitive to long-wavelength light (MmLop1, MmLop2), blue light (MmBLop1) and UV light (MmUVop1) (Figure 6). The four opsins encoded 371, 188, 410 and 380 amino acid residue proteins, respectively. The opsins MmLop1, MmBLop1 and MmUVop1 showed seven transmembrane domains (Appendix A). The three types of opsins indicate that *M. mongolica* eyes have a trichromatic color vision system. Comparisons of the relative proportions of the opsin gene expression reveal that MmLop1 has the highest relative proportion of expression in all the development stages (Figure 6), indicating a high number of green photoreceptors in the compound eyes. 

### 3.6. DEGs Related to Hormone Biosynthesis

Four gene sets related to hormones were revealed in Profiles 3, 4 and 10 (Figure 2). The expression level of DEGs in Profile 3 presented a significant uptrend from N5W to N5R and a downtrend from N5R to A3 (Figure 2). Gene set “steroid hormone receptor activity (GO:0003707)”, which is involved in combination with a steroid hormone and transmitting the signal within the cell to initiate a change in cell activity or function, was enriched in Profile 3. Two gene sets, “insect hormone biosynthesis (map00981)” and “neuropeptide hormone activity (GO:0005184)”, were enriched in Profile 4. Two gene sets, “insect hormone biosynthesis (map00981)” and “terpenoid backbone biosynthesis (map00900)”, were enriched in Profile 10. These results indicate that the drastic change of cell activity and function modulated by hormones occurs not only from N5W to N5R (Profiles 3, 4), but also from N5B to A48 (Profile 10). 

In regard to DEGs involved in the pathway of ecdysteroid biosynthesis, the expression level of four key DEGs presented variations from N5W to A48 (Figure 7). These four DEGs include phm (TRINITY_DN19517_c0_g1) encoding the enzyme ecdysteroid 25-hydroxylase (Phantom; CYP306A1), dib (TRINITY_DN3521_c0_g1) encoding the enzyme ecdysteroid 22-hydroxylase (Disembodied; CYP302A1), sad (TRINITY_DN9020_c0_g1) encoding the enzyme ecdysteroid 2-hydroxylase (Shadow; CYP315A1), and shd (TRINITY_DN24077_c1_g5) encoding the enzyme ecdysteroid 20-hydroxylase (Shade; CYP314A1). The expression levels of these four DEGs presented an uptrend from N5W to N5R and a downtrend from N5R to N5B (Figure 7). Furthermore, the expression level of phm displayed a significant up-regulation from N5W to N5R (Figure 4). The expression level of shd presented a significant down-regulation from N5R to N5B and A3 to A48, respectively (Figure 4). These results indicate that the production of ecdysone and 20-hydroxyedysone (20E) increased dramatically from N5W to N5R and decreased continuously after reaching the peak at N5R.

In regard to the MEKRE93 pathway, the expression of the *Krüppel homolog 1 (Kr-h1)* (TRINITY_DN12498_c0_g1), a key transcription factor in the JH signaling pathway, was highly expressed in N5W but dramatically down-regulated in N5R and N5B, and then displayed a dramatic uptrend after adult emergence (Figure 4 and Figure 7). The decrease and increase of Kr-h1 expression before and after the adult emergence indicate that JH titer was significantly decreased from N5W to N5R and increased after the adult emergence.

### 3.7. DEGs Related to Immunity

Ten gene sets in Profile 4 were revealed to be related to innate immunity for resisting pathogens, of which the expression level of DEGs presented a dramatic downtrend from N5W to N5R (Figure 2). This indicates that the activity of immunity for resisting pathogens was reduced during the period from N5W to N5R. Three gene sets that appeared in Profile 9 are related to the resistance of insecticides and oxidative stress (Figure 2), of which the expression level of DEGs presented a significant uptrend from N5B to A48. These results indicate that oxidative stress responses are essential for *M. mongolica* after the adult emergence.

There were 39 immune-related key genes expressed differently in the eyes from N5W to adult stage, which were classified into three functional groups, including pattern recognition proteins, signal transduction molecules and effector proteins (Appendix A). The 19 pattern-recognition protein genes were assorted into one GNBP1, one PGRP-L, three PGRP-Ss, and 14 βGRPs. The 10 genes encoding signal transduction molecules include one gene encoding Spätzle-processing enzyme (SPE), four genes encoding Spätzle (Spz), three genes encoding Toll, one gene encoding Tollip, and one gene encoding Relish. The 10 genes encoding immune-responsive effectors include five prophenoloxidases (PPO) genes and five antimicrobial peptides (AMPs) genes.

During the period from N5W to N5R, the expression level of 25 immune-related genes were down-regulated and 13 genes were up-regulated, which indicates that activities of all the immune responses show a reduction in this period. During the period between N5R to N5B, 21 immune-related genes were down-regulated, and 15 genes were up-regulated, which indicates that the humoral immune response is partly recovered, including the induction of AMPs regulated by the Toll and Imd pathways. During the period from N5B to A3, 25 immune-related genes were down-regulated, and 11 genes were up-regulated, which indicates that during the molting period, activity of immune responses involved in Toll and Imd pathways decrease and melanization through PPO activating system was increased. During the period from A3 to A48, 16 immune-related genes were down-regulated and 15 genes were up-regulated (for more details see Appendix A).

### 3.8. DEGs Related to Cuticle Development

Gene sets related to cuticle development are revealed in Profiles 3, 4, 6, 8 and 10 (Figure 2). The gene sets related to the chitin metabolic process and structural constituents of chitin-based larval cuticle in Profiles 3, 6 and 8 presented a significant uptrend from N5W to N5R, and the gene set related to structural constituents of cuticle in Profile 4 presented a significant downtrend from N5W to N5R, which indicates that the onset of new cuticle production is in the red-eye period of the 5th-instar nymphal stage. The gene sets in Profile 8 presented a significant uptrend from N5R to N5B, and in Profile 3 remain at a high expression level at N5B. These results indicate that new cuticle was continuously deposited in N5B. The gene sets in Profile 10 presented a significant uptrend from N5B to A3 and a downtrend from A3 to A48. Two gene sets related to cuticle degradation appear in Profile 10, i.e., “chitin catabolic process (GO:0006032)” and “chitinase activity (GO:0004568)”, which could correspond to the degradation and shedding of the old cuticle during adult emergence, i.e., the period between N5B and A3.

There were 62 cuticular protein (CP) genes expressed differently from N5W to the adult stage, including 39, 8, 8, 5 and 2 genes in the RR-2, RR-1, CPAP1, CPAP3 and Tweedle families, respectively (Appendix A), which indicates that the new cuticle of eyes is dominated by cuticular proteins of the RR-2 family. During the period from N5W to N5R, the expression level of five CP genes were down-regulated and 52 CP genes were up-regulated significantly, indicating that new cuticle deposition of eyes was started in this period, i.e., the lens and pseudocone formed in this period. During the period from N5R to N5B, 28 CP genes were up-regulated and 27 CP genes were down-regulated. During the molting period (from N5B to A3), only eight CP genes were up-regulated, and 49 CP genes were down-regulated. During the period from A3 to A48, only seven CP genes were up-regulated (for details see Appendix A).

The chitin-related genes, including one chitin synthetase, one chitin deacetylase and eight chitinase genes were expressed dynamically during the five development stages. During the period from N5W to N5R, seven genes were up-regulated, and three genes were down-regulated. During the period from N5R to N5B, two genes were up-regulated, and five genes were down-regulated. During the period from N5B to A3, five genes were up-regulated, and four genes were down-regulated. During the period from A3 to A48, five genes were up-regulated, and four genes were down-regulated (for details see Appendix A).

### 3.9. Validation of Transcriptome Data Using qRT-PCR

We performed qRT-PCR analysis on the expression of eight genes of *M. mongolica*. They are related to phototransduction cascade (BLop1 and TRPL), cuticle development (CP), hormone biosynthesis (JHAMT), tyrosine metabolism (TH and TDC), innate immunity (PPO), and glucose metabolic (Gld). The trend of the relative expression level of these eight genes were in keeping with the results of RNA-Seq (Appendix A), which confirms that the RNA-Seq data are reliable.

## 4. Discussion

In this study, we reveal how the compound eyes of *M. mongolica* undergo a non-gradual development process based on identification of stage-specific DEGs in the eyes. Our findings are in line with morphological observations demonstrating *M. mongolica* eyes change from white to red and dark brown in the late 5th-instar nymphal stage [18]. Previous morphological observations have also proven that the white eyes of 5th-instar nymphs are composed of a large number of clustered cells, but no ommatidium is visible in the eyes; whereas the red and dark brown eyes of 5th-instar nymphs are composed of ommatidia with eight photoreceptors in each ommatidium, and this is the same in the eyes of adults [18]. In this study, we performed unsupervised hierarchical clustering based on all identified DEGs. The results unveil that dramatic changes of *M. mongolica* eyes occurred in two periods: the first period is between N5W and N5R, and the second is between N5B and A3 (Appendix A). Subsequent cluster analysis of expression profiles performed on all the DEGs further revealed that the tipping point of this eye development is between N5W and N5R, and the critical stage for the eye development is between N5B and A3 (Appendix A).

### 4.1. Dramatic Development of M. mongolica Eyes before Emergence

Chromophore synthesis and the phototransduction cascade are directly related to light sensitivity [1,32]. The results of our present study show that the expression of DEGs related to chromophore synthesis and phototransduction cascade are relatively low in N5W (Figure 5), which indicates that *M. mongolica* nymphs with white eyes have no visual function due to that chromophore status and the molecules related to phototransduction are deficient at this stage. Unexpectedly, the expression level of the opsin gene *MmLop1* at N5W was extremely high when compared with the expression level of the other three opsin genes at this stage, although the expression level of this opsin gene at N5W is the lowest compared with that in other development stages (Figure 6). Previous studies revealed that opsins Rh5 and Rh6 in Johnston’s organ neurons of fruit flies (*Drosophila*) have a mechanotransduction function [33], and chromophore is not essential for the mechanosensory function of opsins [34]. The previous morphological study of *M. mongolica* eyes has demonstrated that sensilla on the eye surface of 5th-instar nymphs disappears in the adults [18]. The results of our study indicate that MmLop1 could play non-visual functions while being related to the mechanosensory role during the white-eye nymphal stage as well as the younger nymphal stages, because the eyes at this stage do not have a usable visual ability (for more reasons see below).

We reveal that the expression levels of all DEGs participating in the phototransduction cascade and six DEGs participating in chromophore synthesis demonstrate show a dramatic uptrend during the period from N5W to N5R (Figure 4 and Figure 5). Most particularly, the expression levels of opsin genes *MmLop1*, *MmLop2*, *MmBLop1* and *MmUVop1* present an intense uptrend from N5W to N5R (Figure 4 and Figure 5). The expression of opsin genes is tightly associated with rhabdom maturation [35]. For example, abnormal expression of the Rh1 opsin gene (i.e., *ninaE*) results in developmental defects of rhabdomeres in *Drosophila* [35] and opsin expression and rhabdom maturation co-occur in honey bees [36]. The results of our study indicate that the formation of the rhabdom starts in the period between N5W and N5R. In addition, the expression level of 52 CP genes presents a significant uptrend in this period, which indicates that the formation of lens and pseudocone also starts between N5W and N5R. Given that ommatidia have been formed in the late 5th-instar nymphal stage (N5B) [18], we conclude that the visual system of *M. mongolica* is gradually completed with a tipping point of structural and functional transformation in the period between N5W and N5R. 

Beyond chromophore synthesis and the phototransduction cascade, accumulation of screening pigments is also critical for the development of the visual function. Screening pigments in pigment cells of compound eyes, such as ommochrome and pteridine, assist in separating ommatidia from each other [7]. For instance, screening pigments in the ommatidia of fruit flies perform two functions, including absorbing stray light to limit light from entering adjacent ommatidia, and promoting the photogeneration of rhodopsins (the light-sensor molecules of rhabdomeres) from metarhodopsins [8]. In the present study, no genes involved in the pteridine biosynthesis pathway were found, and the expression level of genes participating in ommochrome biosynthesis presented significant differences across different developmental stages (Figure 3). These findings indicate that ommochrome is the only screening pigment existing in the pigment cells of ommatidia of *M. mongolica*. The expression level of all four DEGs (*TDO*, *KMO*, *white*, *scarlet*) participating in ommochrome biosynthesis are present at a low level in N5W (Figure 3), but three of them (*TDO*, *KMO*, *scarlet*) see a significant uptrend from N5W to N5R (Figure 4), which argues for ommochrome not accumulating until a tipping point during the period between N5W and N5R. Accumulation of ommochrome and/or pteridine pigments determines the color of insect eyes [37]. For example, the ommochrome pathway converts tryptophan into pigments with various colors ranging from yellow to red, or from brown to black [38]. Therefore, we conclude that it is the accumulation of ommochrome that leads to the color change of compound eyes of *M. mongolica* from white to red and dark brown during the 5th-instar nymphal phase.

In addition, the expression of four key DEGs (i.e., *phm*, *dib*, *sad* and *shd*) participating in ecdysteroid biosynthesis is also detected in compound eyes and shows variations among the five development stages (Figure 7). The expression levels of these four DEGs present an uptrend from N5W to N5R and a downtrend from N5R to N5B (Figure 7). Shd converts ecdysone into 20-hydroxyecdysone (20E) in the peripheral tissues [39,40]. In moths of *Manduca*, *shd* is expressed mainly in the midgut, malpighian tubules, fat body and epidermis and with a very low expression level in the prothoracic gland and nervous system [40]. Moreover, in *Drosophila* flies and *Bombyx* moths, slight expression of phm presents in other tissues other than the prothoracic gland [41]. In the present study, the expression level of *phm* presents a significant up-regulation from N5W to N5R, and *shd* presents a significant down-regulation from N5R to N5B and from A3 to A48 (Figure 4 and Figure 7). These could be related, and indicate that the production of 20E increased dramatically from N5W to N5R and decreased continuously after reaching the peak at N5R. By contrast, the expression of *Kr-h1* decreased dramatically from N5W to N5R, kept a low level at the end of the 5th-instar nymphal stage (N5B), and then continuously increased after the adult emergence (Figure 7). Kr-h1 is a transducer of the anti-metamorphic action of JH involved in the MEKRE93 pathway, which appears to be the status quo action of JH [42]. Previously, observations suggested that the decrease in *Kr-h1* expression at the beginning of the pre-adult stage is critical for metamorphosis, which coincides with the gradual decrease of JH concentration in the haemolymph [42,43,44]. The dynamic change of the expression of *Kr-h1* in *M. mongolica* eyes demonstrates that JH concentration decreases dramatically from N5W to N5R and rapidly increases in the newly emerged adults of *M. mongolica*. These findings, combined with the structural and functional changes of *M. mongolica* eyes before and after the adult emergence, argue that the timing of eye development in this cicada species is closely related to the significant increase of 20E and the decrease of JH during the period between N5W and N5R. In addition, the Toll pathway not only participates in the innate immune response [45], but also mediates the development of embryonic dorsal-ventral patterning [46]. The results showed that the expression level of genes related to the Toll pathway presented a variation from N5W to A48 (Appendix A), which could be related to development function in the compound eyes.

### 4.2. Development of M. mongolica Eyes for Usable Visual Ability after Emergence

Obtaining usable visual ability is vital for cicadas to adapt to their novel above-ground habitat. The expression levels of key DEGs involved in the phototransduction cascade and chromophore synthesis influence light sensitivity directly. In this study, we reveal that eight of the nine DEGs (except for *NinaC*) related to phototransduction cascade present a drastic uptrend from N5B to A3, and four of the eight DEGs related to chromophore synthesis also present a drastic uptrend from N5B to A3 (Figure 4 and Figure 5). The dynamic changes of expression of these key DEGs prove that the light sensitivity of *M. mongolica* eyes is significantly different before and after adult emergence. After the adult emergence, the expression levels of three DEGs (*MmUVop1*, *TRP* and *Arr*) related to the phototransduction cascade and three DEGs encoding PDH related to chromophore synthesis present a significant uptrend from A3 to A48 (Figure 4 and Figure 5). This demonstrates how the period shortly after the adult emergence is critical for functional development of eyes. These results prove that the phototransduction cascade and chromophore synthesis pathway play a positive role allowing *M. mongolica* to adapt to the novel above-ground habitat.

The expression patterns of visual opsin genes are closely related to visual environment [14]. Cicadas are exclusively subterranean during their long nymphal stage but only have a short adult lifespan above ground [17]. According to our field investigation, the adult emergence of *M. mongolica* started at approximately 22:00 p.m. The emerged adults remained on the trunk of the host plant until they could fly away in the morning. Previous studies revealed that, for insects living underground, the dim-light condition is dominated by longer wave-length light [15], and high green sensitivity is linked to the detection of vegetation or the horizon in most insects [47,48]. In this study, we reveal that the opsin expression of *M. mongolica* eyes was dominated by MmLop1 (encoding opsins absorbing long-wavelength light) during both the nymphal and adult stages (Figure 6), which indicates that a high number of green photoreceptors exist in *M. mongolica* eyes.

Insects can exploit the differences in the UV proportion to separate landmarks from background, as with objects reflecting a small amount of UV when compared with the blue or clouded sky [49]. The expression level of *MmUVop1* shows a sharp uptrend from A3 to A48 (Figure 4) which could contribute to *M. mongolica* being able to distinguish objects in diurnal activity after emergence. Furthermore, a previous study suggested that a contrast mechanism involving the UV and green receptors of insect eyes could guarantee a robust separation between natural objects as foreground and sky as background [49]. The expression level of genes encoding green (long-wavelength) light-absorbing opsins (MmLop1, MmLop2) and UV-light-absorbing opsin (MmUVop1) all show a significant uptrend from N5B to A3 (Figure 4) which may be vital for *M. mongolica* obtaining a usable and reliable visual ability after emergence.

Except for opsins, the expression patterns of Ca^+^-permeable light-sensitive transient receptor potential (TRP) and transient receptor potential-like (TRPL) channels are also related to the visual environment [1]. In daylight-active *Drosophila* flies, TRP is the dominate light-sensitive channel in rhabdomeres [50], and the lack of TRPL does not significantly influence *Drosophila* flies’ response to bright light, but their adaptation to low illumination environments is lessened [51]. In contrast, TRPL plays a more significant role in dim light photoreception [1]. For example, TRPL is 10-fold more abundant than TRP in the cockroach *Periplaneta americana* which has apposition eyes and is active in dim light [13]. In our present study, we reveal that TRPL was at least 10-fold more abundant than TRP in all the samples (Figure 6), suggesting TRPL is the dominant ion channel used by *M. mongolica*. The expression level of *TRP* presents a significant up-regulation from A3 to A48 (Figure 4). Furthermore, the relative proportion of TRP increased significantly from A3 to A48, whereas TRPL decreased significantly from A3 to A48 (Figure 6), which may be vital to these cicadas to adapt to the high-illumination habitat above-ground after emergence.

NinaB can be directly used to generate the chromophore 11-cis-3-hydroxyretinal [52]. This chromophore covalently binds to the opsin protein to form Rhodopsin, the light-sensitive molecule [53]. On the whole, expression levels of *NinaB* and opsin genes of *M. mongolica* increased continually from N5W to A48 which should be critical for cicadas to obtain rhodopsin during the development at the end of their nymphal stage and after emergence. The photoreceptor dehydrogenase (PDH) reduces all-trans-3-hydroxyrentinal to all-trans-3-hydroxyretinol in retinal pigment cells (RPCs) and plays a key role in the visual cycle [54]. PDH is functionally equivalent to mammalian RDH, which participates in the chromophore regeneration pathway and is required in humans to prevent retinal degeneration [55]. In Drosophila melanogaster, absence of PDH results in progressive light-dependent loss of rhodopsin and retinal degeneration, and the expression of *RDH12* in PCRs can suppress these defects [54]. It has been demonstrated that the visual cycle allows light-exposed animals to regenerate chromophore and sustain rhodopsin levels under nutrient deprivation conditions, which is critical for maintaining normal visual ability [54]. The expression levels of three PDH genes (i.e., TRINITY_DN30505_c1_g2, TRINITY_DN26316_c0_g6, and TRINITY_DN34823_c0_g1) in *M. mongolica* increased significantly after adult emergence (i.e., from A3 to A48) (Figure 4 and Figure 5), which could be related to prevention of retinal degeneration in the adult stage.

### 4.3. Protective Function of Eye Color Change in Relation to Niche Change

Boulton et al. [56] stated that “the retina represents a paradox, in that, while light and oxygen are essential for vision, these conditions also favor the formation of reactive oxygen species leading to photochemical damage to the retina”. Some light quanta, especially those in the violet and blue regions of the spectrum, carry high energy and present a potential danger for the retinal photoreceptor cells [9]. Prolonged exposure to light is common for many vertebrates and invertebrates. In the optical medium of the eyes of vertebrate animals, the lens plays a key role to protect photoreceptors from ultraviolet and partly from the violet-blue range of visible light [57]. However, the optical medium in compound eyes of insects is transparent to UV [9]. Ommochrome plays an important role in protecting insect retina from photo damage by physically reducing the light intensity reaching the photoreceptors and chemically functioning as an antioxidant [9,10,11]. For example, red-eye mutants of triatomine bugs have fewer ommochromes when compared with the black-eye wild-type, and UV exposure leads to serious damage to ommatidia of the red-eye mutants [10]. In our study, the expression patterns of DEGs participating in ommochrome biosynthesis indicate that there was no ommochrome accumulation in the pigment cells in N5W, and continuous ommochrome accumulation began in the period between N5W and N5R (Figure 3 and Figure 4). These results correspond to the color change of compound eyes from white to red and dark brown at the end of the 5th-instar nymphal stage. In addition, the expression levels of *KMO* and scarlet were significantly up-regulated in A48 when compared with A3 (Figure 3 and Figure 4), which indicates that further production of ommochrome took place in the adult stage. We hypothesize that the accumulation of ommochrome at the end of the 5th-instar nymphal stage and the adult stage is not only a critical process for visual system development but also a secure strategy for the insects to cope with the potential damage caused by intense light after the adult emergence.

Tyrosine metabolism is critical for cuticle tanning of insects and is closely related to the formation of body color [58]. In the present study we reveal that cuticular-secreted protein genes related to the tyrosine metabolism pathway are mainly expressed at the 5th-instar nymphal stage of *M. mongolica*, including genes encoding proteins yellow and enzymes Lac2 (Figure 3). This indicates that the oxidization proteins were secreted into the cuticle before the melanin precursors reached the cuticle. This is consistent with findings in *Papilio xuthus* and *Bombyx mori* [59]. In addition, the epidermal enzyme genes *TH*, *tan* and *ebony* also exhibited a dramatic up-regulated trend from N5B to A3, and *TH* reached the peak at A3 and then exhibited a dramatic downtrend from A3 to A48 (Figure 3 and Figure 4). TH plays an important role in cuticle pigmentation in many insects from several orders, including Diptera [60], Lepidoptera [61], Hemiptera [62], Blattodea [63] and Coleoptera [64,65]. For example, TH together with dopa decarboxylase (DDC) have essential effects on melanization and affect the permeability of the cuticle of cockroaches for coping with desiccation resistance [66]. The results of our present study indicate that the phase shortly after the adult emergence is a critical period for the tanning of cuticles as well as the corneal lens. Given that the production of melanin is related to exposure to oxygen, [67,68] the increase in oxygen concentration after emergence could be a critical factor for the tanning of eyes as well as cuticle. When cicadas emerge from the earth into the novel above-ground habitat, the tanning of cuticle occurring rapidly after emergence would help the adult cicadas cope with potential damage caused by desiccation [66], pathogens [69], photodamage [70], mechanical injury [71], reactive oxygen [72], and high temperature [73].

## 5. Conclusions

In conclusion, we have used transcriptome analyses to clarify the timing and potential molecular mechanisms underlying the structural and functional transformation of eyes of the cicada *M. mongolica*. We have revealed that cicadas have a trichromatic color vision system. The compound eyes undergo a non-gradual developmental process, and dramatic changes of eyes occurred at the end period of the 5th-instar nymphal stage, which is in line with morphological observations. The 5th-instar nymphs with white eyes (i.e., younger 5th-instar nymphs) have no visual function due to ommochrome, chromophore and the molecules related to phototransduction being deficient. It is the accumulation of ommochrome that leads to the color change of their compound eyes from white to red and dark brown during the 5th-instar nymphal stage. Their visual system is completed gradually from a tipping point, which is at the transition phase of the white-eye period and the red-eye period. The drastic structural and functional transformation of their eyes are closely related to the significant decrease of JH and increase of 20E during the pre-adult stage. In this dramatic development process, key DEGs related to chromophore synthesis, the phototransduction cascade and ommochrome biosynthesis play important roles for these insects to obtain visual ability. In addition, the period shortly after the adult emergence is also critical for functional development of their eyes, during which key DEGs related to the phototransduction cascade and chromophore synthesis and the tanning of cuticle, including the corneal lens, play positive roles for these insects to adapt to novel above-ground habitats after emergence. We hypothesize that the accumulation of ommochrome at both the end of 5th-instar nymphal stage and the adult stage is not only a critical process for visual system development but also a secure strategy for the insects to cope with the potential damage caused by intense light and other factors after the adult emergence. These results improve understanding of developmental and adaptive changes in compound eyes in insects which undergo a drastic ontogenic shift in habitat.

## Figures and Tables

**Figure 1 insects-13-01170-f001:**
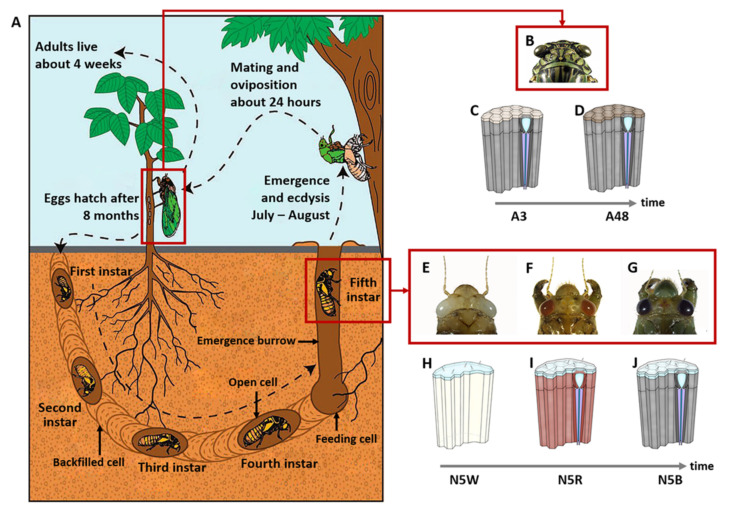
Lifecycle and development of compound eyes of *M. mongolica* during 5th-instar nymphal stage and adult stage (**A**) Lifecycle of *M. mongolica*, adapted with permission from Ref. [19]. (**B**) Head including compound eyes of an adult. (**C**) Ultrastructure of compound eyes of the adult after emergence at 3 h (A3). (**D**) Ultrastructure of compound eyes of the adult after emergence at 48 h (A48). (**E**) Head including white compound eyes of a nymph. (**F**) Head including compound eyes of a nymph with red eyes. (**G**) Head including dark-brown compound eyes of a nymph. (**H**) Structure of a white compound eye in 5th-instar nymphal stage (N5W). (**I**) Structure of a red compound eye in 5th-instar nymphal stage (N5R). (**J**) Structure of a dark-brown compound eye in 5th-instar nymphal stage (N5B).

**Figure 2 insects-13-01170-f002:**
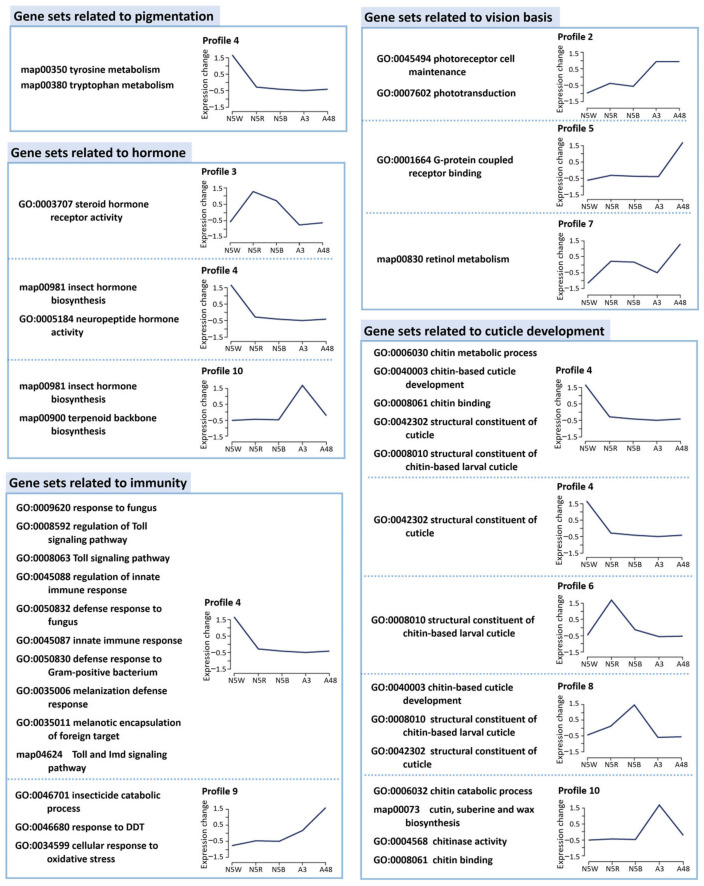
Functional enrichment analysis of profile-specific DEGs (GO terms and KEGG pathways). GO function and KEGG pathway enrichment analyses were performed on DEGs in the 10 profiles, respectively. The significantly-enriched gene sets (*p*-value < 0.01, Enrichment Fold > 5) were focused and analyzed, which are related to specific biological functions including coloration, phototransduction, hormonal regulation, innate immune response and cuticle development, respectively.

**Figure 3 insects-13-01170-f003:**
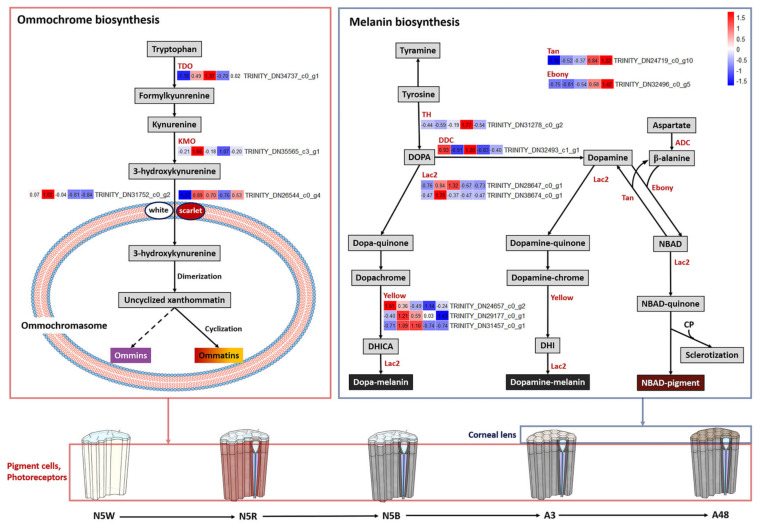
Expression patterns of key DEGs related to pigmentation. Ommochrome biosynthesis pathway is related to the genes encoding tryptophan 2,3-dioxygenase (TDO), kynurenine 3-monooxygenase (KMO), two ABC transporters (white and scarlet). Melanin biosynthesis pathway is related to the genes encoding tyrosine hydroxylase (TH), NBAD hydrolase (Tan), NBAD synthase (ebony), DOPA decarboxylase (DDC), aspartate 1-decarboxylase (ADC), laccase (lac2, two genes), and yellow protein (three genes). The heatmaps based on TPM show the expression levels of the corresponding protein or enzyme genes in five development stages (From left to right, the 5th-instar with white compound eyes (N5W), with red compound eyes (N5R), with dark brown compound eye (N5B), the adults after emergence at 3 h (A3), 48 h (A48).

**Figure 4 insects-13-01170-f004:**
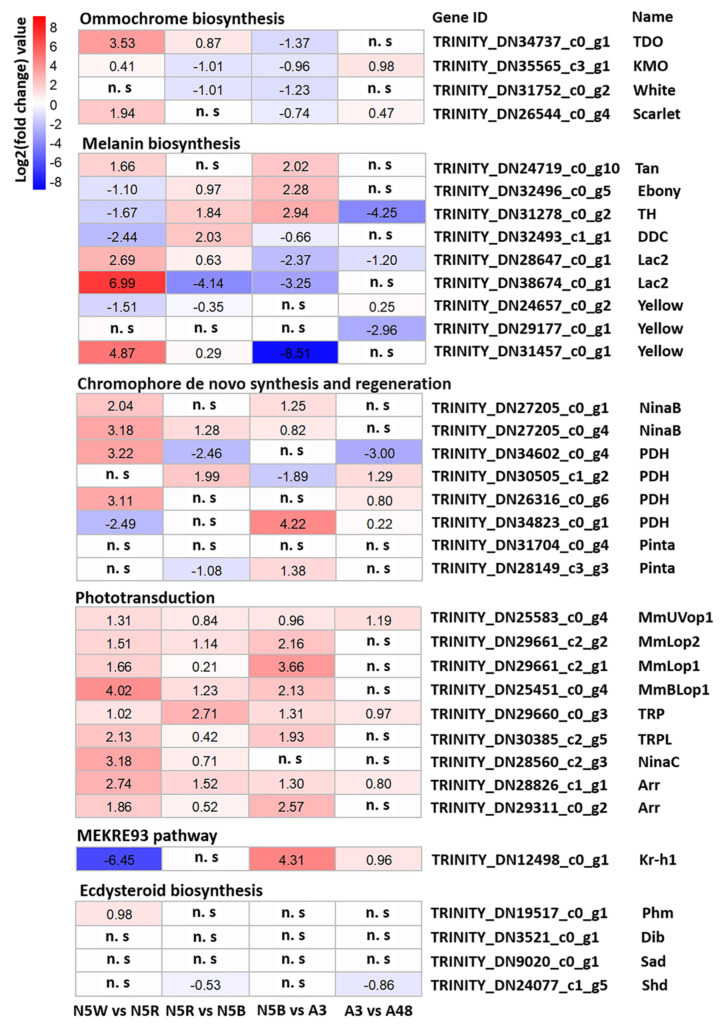
Heatmap of key genes in comparable groups of N5W vs. N5R, N5R vs. N5B, N5B vs. A3, and A3 vs. A48. The log2 fold change-values based on analysis of differentially expressed genes are indicated in the heatmap, which are all at significant level *p*-value < 0.01. No significant differences are marked as “n. s”.

**Figure 5 insects-13-01170-f005:**
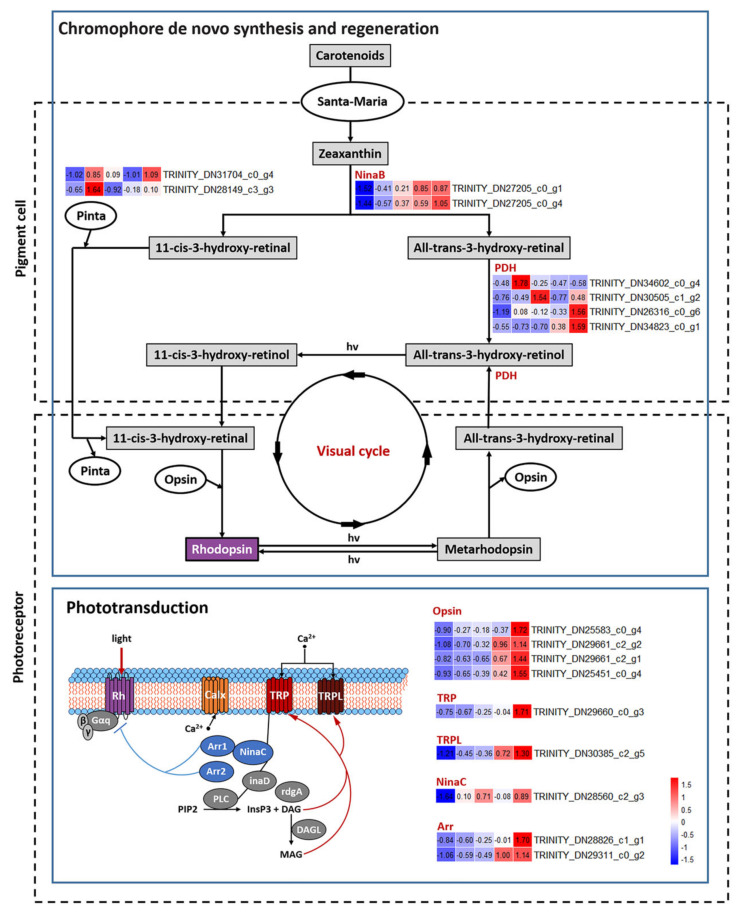
Expression patterns of key DEGs related to structure of visual system. Chromophore de novo synthesis and regeneration is related to the genes encoding retinoid-binding protein (Pinta, two genes (non DEGs)), β, β-carotene-15,15′monooxygenase (NinaB, two genes), photoreceptor dehydrogenase (PDH, four genes). Phototransduction is related to the genes encoding Ca^2+^-permeable light-sensitive transient receptor potential channel (TRP), Ca^2+^-permeable light-sensitive transient receptor potential-like channel (TRPL), myosin III (NinaC), arrestin homologs (two genes), opsins (four genes). The heatmaps based on TPM show the expression levels of the corresponding protein or enzyme genes in five development stages (from left to right, the 5th-instar with white compound eyes (N5W), with red compound eyes (N5R), with dark brown compound eye (N5B), and the adults after emergence at 3 h (A3), 48 h (A48)).

**Figure 6 insects-13-01170-f006:**
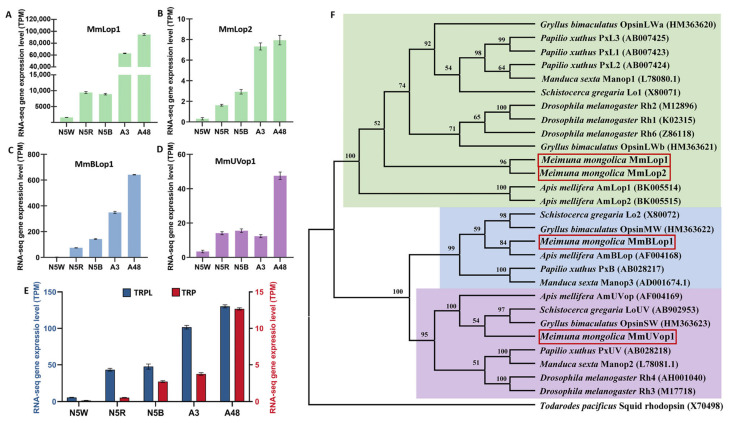
Expression analysis of six genes related to phototransduction and phylogenetic analysis of four opsin genes in *M. mongolica*. (**A**–**E**) RNA-seq gene expression levels of MmLop1, MmLop2, MmBLop1, MmUVop1, TRP and TRPL on five development stages (N5W, N5R, N5B, A3, A48). The 5th-instar with white compound eyes (N5W), with red compound eyes (N5R), with dark brown compound eye (N5B), and the adults after emergence at 3 h (A3), 48 h (A48). (**F**) The phylogenetic tree was based on the maximum likelihood (ML) method. Different opsin nucleotide sequences are grouped into four clusters, which belong to three orders and an outgroup, and are assigned with different colors. Numbers at nodes indicate the bootstrap values. The 29 sequences are indicated by the corresponding scientific names of species followed by their gene names and accession numbers. The sequences of four opsins studied in our study are indicated by solid red rectangular boxes.

**Figure 7 insects-13-01170-f007:**
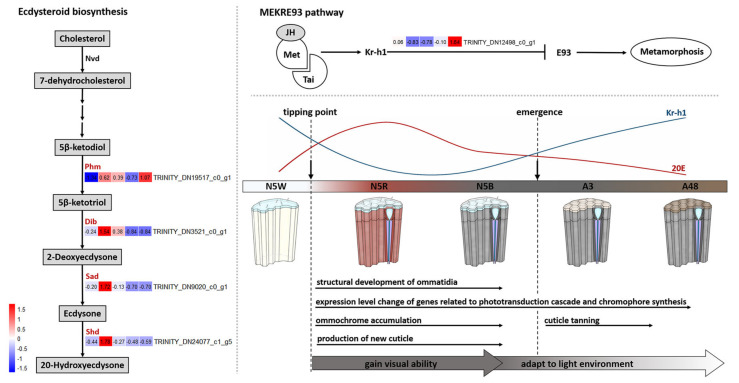
Expression patterns of key DEGs related to hormone biosynthesis. Ecdysteroid biosynthesis pathway is related to the genes encoding ecdysteroid 25-hydroxylase (phm), ecdysteroid 22-hydroxylase (dib), and ecdysteroid 2-hydroxylase (sad), ecdysteroid 20-hydroxylase (shd). MEKRE93 pathway is related to the gene encoding the Krüppel homolog 1 (Kr-h1). The heatmaps based on TPM show the expression levels of the corresponding protein or enzyme genes in five development stages (from left to right, the 5th-instar with white compound eyes (N5W), with red compound eyes (N5R), with dark brown compound eyes (N5B), and the adults after emergence at 3 h (A3), 48 h (A48)).

## Data Availability

Data is contained within the article and Appendix A. The RNA-Seq data has been submitted to the Sequence Read Archive (SRA) of the National Center for Biotechnology Information under BioProject number PRJNA888061.

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
