# Peer review of "A Non-Gradual Development Process of Cicada Eyes at the End of the Fifth-Instar Nymphal Stage to Obtain Visual Ability"

_insects, 2022, doi:10.3390/insects13121170_

Round 1

Reviewer 1 Report

Overall the manuscript use transcriptome analyses to clarify the timing and potential molecular mechanisms underlying the structural and functional transformation of the eyes of the cicada Meimuna mongolica. However, some important details from the Material and Methods and Results sections of the MS are missing (see Major Remarks). I would like to see the authors address the comments below before it can be published. I divide my comments into major and minor remarks below:

Major remarks

Line 35: In the introduction, there are too many descriptions of compound eye structure and life history, but relatively few descriptions of DEG or ommochrome. For example, there no introduceof tryptophan metabolism and tyrosine metabolism relate to pigment biosynthesis in pigment cells of compound eyes and melanization of cuticle.

Line 106: Why combine two individuals as one biological sample? Why not other quantities? Please indicate the basis of combine two individuals.

Line 252-253. This study mainly focuses on the vision of the Meimuna mongolica, but why does the content include the analysis of “hormonal regulation, internal immune response and cuticle development”?

Minor issues.

Line 16: Insects’ visual system reflects high adaptation to their habitats. Is the same to the first sentence of simple summary (Line 8).

Line 16: All the Cicadas have the eye color change?

Line 25: The DEGs should not be abbreviated in the fire time.

Line 36: “invertebrates” change to “animals”.

Line 39: delete “in their world of competition and selection”

Line 40 “have evolved as ”change to “is”.

Line 42  delete “anatomical”

Line 47- 48. change to “Compound eyes in different insects are vary widely in resolving power and light sensitivity.”

Reviewer 2 Report

This manuscript reports the findings from a transcriptomic analysis of eye development in a cicada species. The study nicely builds on the previous structural description of compound eye development during metamorphosis of the same species. The experimental design is well conceived and the question how the development of the compound eyes unfolds at the level of gene pathway regulation is an interesting and significant one. Data generation, documentation, and presentation meet state-of-the-art standards. Overall, the manuscript is written in a well understandable manner and easy to follow.

There are some major and minor issues to address for the manuscript to reach its full potential for the wide readership it deserves.

Major issues:

1.

It is essential to add a figure to the introduction that summarizes the life cycle of Meimuna mongolica and indicates the crucial steps of compound eye development so that the reader is well prepared for the results part.

2.

There are 2 LW opsins. That leads to the question of whether Meimuna mongolica has ocelli and whether one of them might be specific for the compound eyes while the other one could be specific for the ocelli. The ocelli-specific one would be characterized by much lower RPKM values. Ideally, of course, this could be tested by ocellus-specific RT-PCR experiments.

Minor issues:

1.

“Aboveground”

Above ground

2.

“The compound eyes (in- 104

cluding lens, retinal and optic lobes)”

retinal = retinal tissue or retina

3.

“Trizol reagent (Invitrogen, CA, USA) was used to the extraction…”

FOR the extraction

4.

“Analyses of the total RNA quantity and purity were 112

based on Bioanalyzer 2100 and RNA 1000 Nano LabChip Kit (Agilent, CA, USA) with an 113

RIN number >7.0. Poly(A) RNA was purified from total RNA (5 ug) using poly-T oligo- 114

attached magnetic beads using two rounds of purification.”

Break up sentence. It is close to impossible to understand.

5.

“To remove the reads containing adaptor contamination, low quality bases and unde- 123

termined bases, cutadapt [12] and perl scripts in house were used.”

Make perl scripts available in supp data

6.

“The four opsin genes were aligned with corresponding genes of related species to 156

ensure the reliability of the four genes as the homologs of opsin genes. All selected nucle- 157

otide sequences were aligned using the ClustalW tool before phylogenetic analysis. Phy- 158

logenetic analyses are based on 29 nucleotide sequences, including the four opsin genes 159

from M. mongolica identified in this study, 24 opsin genes from seven other insect species 160

and one opsin gene from the squid Todarodes pacificus.”

Protein sequence analysis would be more informative. But the results are clear enough to keep them as they are. However, it would be helpful if a multiple sequence alignment of the protein sequences could be added to the supplementary data.

7.

“The RNA-Seq has been submitted to the Sequence Read Archive (SRA) of the Na- 188

tional Center for Biotechnology Information under BioProject number PRJNA888061. The 189

release date is 2022-10-14. 190”

Specify data. Does that include the transcript assemblies? If not, I would submit them as well.

8.

“The depth of sequencing we carried out is >6 GB per mixed sample (including 2 in- 193

dividuals for each repetition of the five sampled stages, i.e., N5W, N5R, N5B, A3, A5) to 194

ensure coverage of the M. mongolica transcriptomes.”

I highly recommend using past tense consistently throughout the entire results part.

9.

“3.1 Stage-specific RNA-seq of compound eyes of M. mongolica”

This paragraph reads like a method description. It does not describe findings that answer any specific question. I think it can be moved to/integrated with the results section.

10.

“3.4 DEGs related to coloration”

coloration = ye pigmentation?

11.

“...the expression level of KMO and scarlet were significantly…”

levelS

12.

“ The gene (TRIN- 307

ITY_DN28647_c0_g1) encoding the enzyme laccase (Lac2) was highly-expressed in N5B, 308

and the other gene (TRINITY_DN38674_c0_g1) encoding Lac2 was highly-expressed in 309

N5R. In addition, the epidermal enzyme gene DDC (TRINITY_DN32493_c1_g1) encod- 310

ing DOPA decarboxylase was highly-expressed in N5W and N5B (Figure 2).”

Duplicated Laccase gene?

13.

“ The expression patterns of the above-mentioned genes, in particular 318

TH, indicate that the eyes were tanned mainly after adult emergence and that the 319

melanization of eyes did not occur at the 5th-instar nymphal stage.”

Tanning is also also result of exposure to oxygen

14.

“This 326

indicates that the adult emergence period, in addition to the period between N5W and 327

N5R, is a critical point in time for eye development and functional transformation.”

Terminal differentiation is probably a better fitting description for this stage of retinal development.

15.

“3.5 DEGs related to visual system development”

The genes described in the paragraph are structural vision genes not developmental regulators.

16.

“sion level of four key DEGs presents variations from N5W to A48 (Figure 6). These four 426

DEGs include phm (TRINITY_DN19517_c0_g1) encoding the enzyme ecdysteroid 25- 427

hydroxylase (Phantom; CYP306A1), dib (TRINITY_DN3521_c0_g1) encoding the enzyme 428

ecdysteroid 22-hydroxylase (Disembodied; CYP302A1), sad (TRINITY_DN9020_c0_g1) 429

encoding the enzyme ecdysteroid 2-hydroxylase (Shadow; CYP315A1), and shd (TRIN- 430

ITY_DN24077_c1_g5) encoding the enzyme ecdysteroid 20-hydroxylase (Shade; 431

CYP314A1). The expression levels of these four DEGs present an uptrend from N5W to 432”

The results parts would be MUCH easier to read if only the gene names are provided in the text. The transcript IDs should be/could be provided in a supplementary table.

17.

“3.7 DEGs related to immunity”

Note that some of these genes also function as developmental regulators. So their expression may be more due to developmental functions instead of immunity issues.
